# The DASciS Software for BSI Calculation as a Valuable Prognostic Tool in mCRPC Treated with 223RaCl2: A Multicenter Italian Study

**DOI:** 10.3390/biomedicines11041103

**Published:** 2023-04-05

**Authors:** Maria Silvia De Feo, Viviana Frantellizzi, Matteo Bauckneht, Alessio Farcomeni, Luca Filippi, Elisa Lodi Rizzini, Valentina Lavelli, Maria Lina Stazza, Tania Di Raimondo, Giuseppe Fornarini, Sara Elena Rebuzzi, Mammini Filippo, Paolo Mammucci, Andrea Marongiu, Fabio Monari, Giuseppe Rubini, Angela Spanu, Giuseppe De Vincentis

**Affiliations:** 1Department of Radiological Sciences, Oncology and Anatomo-Pathology, Sapienza, University of Rome, 00161 Rome, Italy; 2Department of Health Sciences (DISSAL), University of Genova, 16132 Genova, Italy; 3Nuclear Medicine, IRCCS Ospedale Policlinico San Martino, 16132 Genova, Italy; 4Department of Economics & Finance, University of Rome “Tor Vergata”, 00133 Rome, Italy; 5Department of Nuclear Medicine, Santa Maria Goretti Hospital, 04100 Latina, Italy; 6Radiation Oncology, IRCSS Azienza Ospedaliero-Universitaria di Bologna, 40138 Bologna, Italy; 7Nuclear Medicine Section, Interdisciplinary Department of Medicine, University of Bari “Aldo Moro”, 70124 Bari, Italy; 8Unit of Nuclear Medicine, Department of Medicine, Surgery and Pharmacy, University of Sassari, 07100 Sassari, Italy; 9Medical Oncology Unit 1, IRCCS Ospedale Policlinico San Martino, 16132 Genova, Italy; 10Medical Oncology, Ospedale San Paolo, 17100 Savona, Italy; 11Department of Internal Medicine and Medical Specialties (Di.M.I.), University of Genova, 16132 Genova, Italy; 12Radiation Oncology, Department of Medical and Surgical Siences (DIMEC), Alma Mater Studiorum Bologna University, 40138 Bologna, Italy

**Keywords:** mCRPC, DASciS software, bone scan index, radium-223, overall survival, bone metastasis, bone scintigraphy

## Abstract

Background/Aim: Radium-223 dichloride (^223^RaCl_2_) represents a therapeutic option for metastatic castration-resistant prostate cancer (mCRPC) patients dealing with symptomatic bone metastases. The identification of baseline variables potentially affecting the life-prolonging role of ^223^RaCl_2_ is still ongoing. Bone scan index (BSI) defines the total load of bone metastatic disease detected on a bone scan (BS) and is expressed as a percentage value of the whole bone mass. The aim of this multicenter study was to assess the impact of baseline BSI on overall survival (OS) in mCRPC patients treated with ^223^RaCl_2_. For this purpose, the DASciS software developed by the Sapienza University of Rome for BSI calculation was shared between six Italian Nuclear Medicine Units. Methods: 370 pre-treatment BS were analyzed through the DASciS software. Other clinical variables relevant to OS analysis were taken into account for the statistical analysis. Results: Of a total of 370 patients, 326 subjects had died at the time of our retrospective analysis. The median OS time from the first cycle of ^223^RaCl_2_ to the date of death from any cause or last contact was 13 months (95%CI 12–14 months). The mean BSI value resulted in 2.98% ± 2.42. The center-adjusted univariate analysis showed that baseline BSI was significantly associated with OS as an independent risk factor (HR 1.137, 95%CI: 1.052–1.230, *p* = 0.001), meaning that patients with higher BSI values had worse OS. When adjusting for other measures on multivariate analysis, in addition to Gleason score and baseline values of Hb, tALP, and PSA, baseline BSI was confirmed to be a statistically significant parameter (HR 1.054, 95%CI: 1.040–1.068, *p* < 0.001). Conclusions: Baseline BSI significantly predicts OS in mCRPC treated with ^223^RaCl_2_. The DASciS software was revealed to be a valuable tool for BSI calculation, showing rapid processing time and requiring no more than a single demonstrative training for each participating center.

## 1. Introduction

### 1.1. Radium-223 Dichloride (^223^RaCl_2_)

Bone metastatic involvement represents the end-stage of the disease for many patients with metastatic castration-resistant prostate cancer (mCRPC) [1]. Dealing with marrow failure, impaired mobility, pathologic fractures, disabling bone pain, and spinal cord or nerve root compression, these subjects experience a significant decline in their clinical condition and have poor overall survival (OS) [2]. Radium-223 dichloride (^223^RaCl_2_; Xofigo^®^; Bayer HealthCare Pharmaceuticals Inc., Hanover, NJ, USA) is a therapeutic calcium-mimetic agent binding areas of increased osteoblastic activity, including bone metastases [3]. The high-energy alpha particles resulting from its decay (physical half-life of 11.4 days) induce predominantly nonrepairable double-stranded DNA breaks in a short range of less than 100 μm [4], leading to specific cancer cell targeting with minimal hematological adverse events [5,6,7]. ^223^RaCl_2_ was approved by the Food and Drug Administration (FDA) in 2013 [8] and rapidly introduced in the clinical setting as a therapeutic option for mCRPC with symptomatic bone metastases and no evidence of visceral metastatic involvement [9] after the phase III clinical trial ALSYMPCA showed a palliative effect on bone pain, delayed time to first symptomatic skeletal-related events (SRE) and significant improvement of OS [10]. Although the role of ^223^RaCl_2_ in prolonging OS is well established, clinical practice has reported lower survival benefits than the 3.6 months of the ALSYMPCA study, maybe due to the suboptimal selection of patients with unfavorable prognostic factors [9,11]. In addition, a formal warning promoted by the European Medicines Agency (EMA) in 2018, moved ^223^RaCl_2_ treatment to the advanced stages of the disease, limiting the prescription to mCRPC patients having more than six osteoblastic lesions at bone scan (BS) and pre-treated with at least two systemic therapies or ineligible for any systemic treatments. EMA restricts the use of prostate cancer medicine XOFIGO [12]. In this scenario, the patient selection process results were extremely challenging, and identifying reliable prognostic factors represents a crucial clinical issue worth continuous research.

### 1.2. Bone Scan Index (BSI)

BS with metastable technetium-99(^99m^Tc)-labeled diphosphonates has been used since the early 1970s and has over time become the functional imaging of choice for the management of bone metastases in routine clinical practice [13]. Despite BS interpretation being essentially qualitative and observer-dependent, in the last years the development of artificial intelligence (AI) has opened new fascinating horizons and alternative methods have been proposed. The bone scan index (BSI) defines the total load of bone metastatic disease detected on BS and is expressed as a percentage value of the whole bone mass [14]. This methodological approach allows the expression of BS data as a single quantitative measure, particularly useful to evaluate the burden of bone metastatic involvement. Lowering the time required for manual selection of all sites of increased tracer fixation, as well as providing reproducible and reliable data, are the objectives of computer-based algorithms specifically developed for the automatic selection of metastatic areas on BS.

### 1.3. DASciS Software for BSI Calculation

The BONENAVI^®^ software (Japan), as well as the EXINI bone software package (EXINI Diagnostics, Lund, Sweden), have been specifically developed for BSI calculation, enabling the use of this prognostic measure in prostate cancer patients [15,16]. A few years ago, an engineering team at the Sapienza University of Rome, developed a specific program for BSI calculation, the DASciS software, which has been validated in 2019 in a monocentric analysis involving 127 mCRPC patients treated with ^223^RaCl_2_ and proposed BSI as a reliable prognostic factor [17].

### 1.4. Aim of the Study

Based on these preliminary results, we conducted a multicentric retrospective study with the aim to corroborate the prognostic relevance of baseline BSI calculated through the DASciS software for mCRPC patients treated with ^223^RaCl_2_. For this purpose, the DASciS software was shared between six different Italian Nuclear Medicine Units.

## 2. Materials and Methods

The study was approved by the local Ethical Committee of each adhering center and was performed in accordance with the ethical standards of the 1964 Declaration of Helsinki and its later amendments. All patients signed a written Informed Consent, which included the use of anonymized data for retrospective research purposes, before each ^223^RaCl_2_ administration. The present retrospective multicenter study included 370 consecutive mCRPC patients eligible for ^223^RaCl_2_ therapy according to the criteria in force at the time of enrollment [12,18,19,20] and treated in six Italian Nuclear Medicine Units between July 2015 to November 2022, time of the analysis. Conforming to the standard selection criteria for ^223^RaCl_2_ treatment, all enrolled patients had a diagnosis of mCRPC with symptomatic bone metastases, no visceral metastatic involvement except for malignant lymphadenopathies with less than 3 cm in the short-axis diameter assessed through a Computed Tomography (CT) scan and/or a Positron Emission Tomography (PET)-CT scan performed before enrollment, adequate hematological, hepatic and renal function [21] and absence of inflammatory bowel disease (IBD). The treatment schedule consists of an intravenous injection of 55KBq/Kg of body weight, dispensed every 28 days, for a total of 6 cycles [22]. Having received at least one cycle of radionuclide therapy was required for enrollment in the present study. During treatment, patients continued androgen deprivation therapy (ADT), while either chemotherapy, abiraterone, or enzalutamide were discontinued before the first ^223^RaCl_2_ administration. Conventional analgesics and glucocorticoids were administered to control pain, as prescribed by the best standard of care and eventual toxicities were managed according to current guidelines. The unavailability of the baseline ^99m^Tc-hydroxydiphosphonate (HDP) BS represented an exclusion criterion, as it was essential for calculating the baseline BSI. BSI data obtained from images acquired before treatment were analyzed from an OS prediction’s perspective. OS was established from the date of the first administration of ^223^RaCl_2_ until the date of death from any cause or last contact (last ^223^RaCl_2_ administration, last follow-up phone call, or last follow-up BS). Other baseline clinical variables relevant for OS analysis specifically age, Gleason Score, Eastern Cooperative Oncology Group (ECOG) performance status (PS) score, previous primary treatment, presence of lymphadenopathies, prior chemotherapy, use of bisphosphonates/denosumab, number of previous treatments, number of cycles of ^223^RaCl_2_ received, baseline values of hemoglobin (Hb), total alkaline phosphatase (tALP) and prostate-specific antigen (PSA), were independently collected by the six involved centers and then shared, put together and taken into account for the statistical analysis. In all involved centers, scintigraphic images were acquired approximately 2 h after the intravenous injection of 300–740 MBq of ^99m^Tc-HDP [23] providing a total-body image in anterior/posterior projection (matrix size was 1024 × 256, energy peak centered at 140 KeV +/− 10%). In some cases, images of particular anatomical regions were added. The DASciS software developed by the engineering team from the Sapienza University of Rome was shared with all involved Nuclear Medicine Units. After online training, the six centers independently processed the baseline BS images of their patients with the DASciS software. BSI values were independently collected and then put together for statistical analysis. 

### 2.1. DASciS Software

DASciS software, which stands for Dicom Analyzer Scintigraphy Software, is an automatic tool for BS quantitation. Gamma Cameras usually output images in DICOM format. The output file contains the actual images together with all the metadata gathered during the exam. Through DASciS software, we can visually analyze those files, computing the areas relative to the ill portions of the patient’s skeleton. The software performs the computation based on the intensity of the pixels. More specifically, once the operator has selected a pixel on the image that has been recognized as a portion of the ill skeleton, the software automatically selects all the pixels in the image whose intensity is equal to or higher than the picked one. All those pixels are clustered into, potentially, multiple regions of interest (ROIs). The operator can also manually exclude some ROIs that correspond to areas of increased fixation not related to metastatic pathology (such as recent or previous fractures and the bladder) and improve the automated output by manually changing the contrast intensity in a limited range. Once the operator has successfully analyzed the file, DASciS outputs a file containing the statistics and the relevant metadata of the investigation. More specifically, the statistics include the cumulative percentage of ill regions computed with respect to the total image area of the patient. The metadata, instead, contains patient generalities and the date of acquisition. From a more technical point of view, the software has been developed in Java to grant cross-compatibility with all the most used Operating Systems (Windows, MacOS, Linux) and to easily prototype an effective Graphical User Interface (GUI). Image processing is performed using the well-known OpenCV library [24] while DICOM files are handled using the open-source PixelMed library. The OpenCV library implements several efficient Computer Vision algorithms, similar to the ones used in DASciS to perform intensity-based clustering of pixels—based on the well-known approach of Suzuki [25] and to calculate the cluster area—through Green’s theorem [26]. Finally, the program saves the processed statistical data in a file (CSV or other extensions). To summarize, the DASciS software semi-automatically identifies all the areas of increased fixation of bone-targeted radiotracer only requiring the operator to identify one of these areas. With this method is possible to make a quantitative analysis of the ROIs representing the metastatic bone towards the whole-body bone mass, obtaining a percentage of the bone metastatic load. The reproducibility of this method had been previously examined by comparing results obtained by three independent, blinded operators, with different degrees of expertise in nuclear medicine techniques.

### 2.2. Statistical Analysis

Data are expressed as mean ± standard deviation, or median ± MAD where appropriate. Quantiles of survival were estimated through the Kaplan-Meier product limit estimator. The relationship between baseline covariates and the time-to-event endpoint was assessed by means of univariate and multivariable Cox regression models, adjusted for possible center-specific effects. Robust standard errors were computed accordingly. The final multivariable model was selected using a forward stepwise procedure based on the Akaike Information Criterion (AIC). A sensitivity analysis was also performed, showing robustness to the criterion used for model selection. The threshold for statistical significance was established at 5% before the analysis. All analyses are performed with the R software version 4.1.2.

## 3. Results

A total of 370 pre-treatment BS have been analyzed through the DASciS program to calculate the BSI and assess its association with OS in mCRPC patients treated with ^223^RaCl_2_. Baseline patients’ characteristics are shown in Table 1. The mean age was 73.6 ± 8.1 years. The vast majority of subjects presented in the involved Nuclear Medicine Units with an ECOG-PS of 0 or 1 (83%). According to histological data, the mean Gleason score was 7.9 ± 1.0. Only 43% of patients underwent either radical prostatectomy or radiotherapy, while the remaining percentage of cases did not receive any type of primary treatment. The largest part had undergone at least one cycle of chemotherapy with a mean number of treatment lines prior to ^223^RaCl_2_ of 1.9 ± 1.4. Approximately one-third of patients had known lymphadenopathies at the time of presentation. Just over half of patients (52.2%) made concomitant use of bisphosphonates/denosumab at the time of enrollment. In the total cohort, more than 70% had completed all six scheduled administrations, while only a minority of subjects had received only one cycle of treatment. The mean baseline values of Hb, tALP, and PSA were 11.9 ± 1.6 g/dL, 251.1 ± 310.9 U/L, and 230.7 ± 580.1 ng/mL, respectively. As concerning bone metastatic involvement, the BSI was calculated with the DASciS program on pre-treatment BS (Figure 1). The mean BSI value resulted in 2.98% ± 2.42. Of 370 patients, 326 subjects had died at the time of our analysis. The median OS time was 13 months (95%CI 12–14 months), as shown in Figure 2. The results of the center-adjusted univariate analysis are shown in Table 2. Considering clinical covariates in univariate models, several clinical aspects showed an impact on OS. Baseline BSI was significantly associated with OS as an independent risk factor (HR 1.137, 95%CI: 1.052–1.230, *p* = 0.001), meaning that higher values are predictors of worse OS. Concerning the other clinical variables exhibiting a significant association with OS, higher age, Gleason score, ECOG-PS, tALP, and PSA, as well as the presence of lymphadenopathies at the time of enrollment, resulted independently associated with an increased risk of death, while previous primary treatment (radical prostatectomy/radiotherapy) and higher baseline Hb values significantly associated with better outcomes. When adjusting for other measures on multivariate analysis, the Gleason score and baseline values of BSI, Hb, tALP, and PSA were confirmed to be statistically significant parameters. This means that for the same Gleason score and the same baseline values of Hb, tALP, and PSA, there is an effect of BSI on OS, indicating that patients with lower values of bone metastatic involvement have longer OS (HR 1.054, 95%CI: 1.040–1.068, *p* < 0.001). The results of the center-adjusted multivariate analysis are shown in detail in Table 3.

## 4. Discussion

OS gain represents the main distinctive feature between ^223^RaCl_2_ and other palliative bone-targeting therapies [27]. According to the registrative phase III clinical trial ALSYMPCA, six cycles of ^223^RaCl_2_ resulted in a 30% reduction in risk of death compared to placebo, with a reported median OS of 14.9 and 11.3 months in the experimental arm and in the control arm respectively [10]. However, in the real-life setting, several clinical variables have a non-negligible importance in determining the role of ^223^RaCl_2_ as the life-prolonging agent. Different prognostic factors, ranging from baseline values of Hb, PSA, and tALP, to the evaluation of the number of prior systemic treatments, ECOG-PS, and quality of life, to name a few, have been proposed [9,11,28]. Multidimensional approaches taking into account different baseline variables were demonstrated to have higher prognostic relevance. A three-variable prognostic score taking into account baseline patients’ Hb, ECOG-PS, and PSA has been proposed [29] and recently further validated in a multicentric study [30]. Similarly, a composite prognostic score including inflammatory indices from peripheral blood and clinical factors (ECOG-PS, tALP, and the number of bone metastases at the bone scan) has been validated in this clinical setting [19,20]. In such background, the identification of additional reliable prognostic factors, able to select patients most likely to benefit from ^223^RaCl_2_, still results of great importance. Beyond the enrollment procedure for ^223^RaCl_2_ therapy, a simple count of the number of bone metastases turns out to be extremely reductive. Indeed, a number of studies focused on the use of imaging semi-quantification as a tool to assess tumor spread in mCRPC candidates to receive ^223^RaCl_2_ therapy [31,32,33,34,35,36,37]. The BSI, a measure expressing the fraction of the skeleton involved by the tumor as a percentage value of the whole body mass [14], has been proposed for determining the extent of disease, monitoring disease progression, and assessing treatment response in different malignancies. This measure has proven to be a valuable prognostic marker in patients with prostate cancer, allowing the identification of subjects with distinct prognoses for better stratification in clinical trials, as its prognostic significance is independent of treatment choice [38,39]. However, the application of this metric is hampered by the tedium of manual calculation. During the last few years, computerized BSI calculation has greatly extended the potential for rapid, quantitative analysis of planar BS by outperforming manual approaches in terms of reproducibility and especially speed. The BSI calculated through the BONENAVI^®^ software, developed in Japan and not commercially available in Western countries, has been validated in Japan as OS predictor in mCRPC patients treated with enzalutamide and docetaxel [40,41], prospected as a promising tool for the assessment of treatment response to ^223^RaCl_2_ in a small cohort of patients [42], and recently included in a novel nomogram for the prognostic evaluation of patients undergoing ^223^RaCl_2_ [42]. Similarly, the EXINI bone software package (EXINI Diagnostics, Lund, Sweden) [43,44], has been used to demonstrate the role of BSI as a promising biomarker for prognostication of OS and hematologic toxicity in late-stage mCRPC patients receiving ^223^RaCl_2_ [32] and subsequently to assess radiographic response to ^223^RaCl_2_ treatment in a multicentric analysis [45]. The present Italian multicentric study involved six Nuclear Medicine Units sharing a specific software developed by an engineering team of the Sapienza University of Rome for BSI calculation, the DASciS software. The baseline BS of 370 mCRPC patients treated with ^223^RaCl_2_ was retrospectively analyzed from an OS perspective. The study confirmed the preliminary results of a previous monocentric analysis [17], showing how BSI can be considered a predictor of OS in mCRPC treated with ^223^RaCl_2_. Our results are in line with data reported in the abovementioned papers and obtained by using different software for reliable BSI calculation [41,42,46]. In the univariate analysis, BSI was demonstrated to be significantly associated with OS as an independent risk factor, meaning that patients with higher values of bone metastatic involvement have worse survival outcomes. The multivariate analysis confirmed the prognostic relevance of BSI, showing that among patients with the same clinical condition in terms of Gleason score and baseline values of Hb, tALP, and PSA, those with lower BSI, are more suitable to obtain a greater survival benefit from ^223^RaCl_2_. As concerning other baseline variables, the results of the univariate analysis confirmed how previous primary radical treatment represents a protective factor in this cohort of patients, in line with a multicentric study published in 2019 [47]. On the contrary, concomitant therapy with bisphosphonate such as zoledronic acid and the RANK ligand (RANKL) inhibitor Denosumab revealed no impact on OS, as previously reported [48]. According to most of the literature data, the median OS resulted inferior to the 14.9 months reported in the ALSYMPCA study, being of 13 months in our cohort [10,49]. It is worth underlining that in the multivariate analysis, the BSI showed better performance than the ECOG-PS, a well-known reliable tool for prognostic assessment in mCRPC patients undergoing ^223^RaCl_2_, according to consistent data from the literature [19,30,35]. Based on these considerations and due to its valuable utility in patients’ stratification, independently from treatment choice, baseline BSI could be investigated as an additional variable to be included in a multidimensional approach taking into account different data, such as baseline Hb, tALP and PSA which corroborated their prognostic relevance in the present analysis. A non-negligible part of the intrinsic value of BSI lies in the fact that despite other imaging modalities have been proven to be superior [50,51], low costs, ease of use, and wide availability are reasons for preserving BS’s importance in the management of bone metastasis, with the potential of further improving through automated quantification programs. Through the DASciS software, a complex parameter such as the burden of bone metastatic disease was shared between all involved centers as simple numerical data allowing for the evaluation of a consistent number of patients. The program was also demonstrated to be a simple tool, requiring no more than a single demonstrative training for each participating center. After gaining experience, the estimated time for a single BSI calculation is less than one minute, significantly much lower than the time requested for manual BSI calculation. Considering that ^99m^Tc-labelled diphosphonates are nonspecific markers of osteoblastic activity and increased uptake can be observed in previous fractures, Paget’s disease, degenerative joint diseases, as well as in case of inflammation and trauma, the possibility of manually excluding from BSI calculation, ROIs corresponding to areas of increased fixation not related to metastatic pathology, makes the DASciS software a precious instrument in the hands of nuclear medicine physicians, but not presuming to replace expert assessment.

## 5. Limitations

A mention of some major limitations and drawbacks is needed. First of all, it is worth noting that the BSI calculation, independent of whether manual or software-based, incorporates the intrinsic limitations of BS, most notably, it reflects osteoblastic activity and not the tumor itself. Moreover, even if the specific function of the DASciS software allows the exclusion of areas of increased uptake not related to metastatic involvement from BSI calculation, mainly represents a non-trascurabile added value of this program, it leaves room for interpretation errors. In addition, BSI changes during treatment have not been evaluated in the present analysis, only focusing on the prognostic value of baseline BSI, reflecting patients’ condition at the time of enrollment. Furthermore, our population includes subjects satisfying heterogeneous recruitment criteria, being enrolled from 2015 to 2022. To conclude, despite the high number of included patients, the retrospective nature of this study represents a possible limitation, thus, it would be useful to perform a larger-scale prospective trial to validate our results.

## 6. Conclusions

The present Italian multicentric study confirmed the prognostic relevance of baseline BSI in mCRPC treated with ^223^RaCl_2_ in a large cohort of patients. The DASciS software was revealed to be a valuable tool providing reliable measurements of bone metastatic involvement, requiring minimal training of Nuclear Medicine physicians, showing high reproducibility, and having rapid processing time. Such automated BSI calculation might be added to other clinical variables to provide a more accurate prognostic assessment of this patient population. Future studies could focus on the ability of this quantitative computer-based algorithm to accurately detect changes in skeletal tumor burden over time, aiming to assess patients’ response to ^223^RaCl_2_ therapy.

## Figures and Tables

**Figure 1 biomedicines-11-01103-f001:**
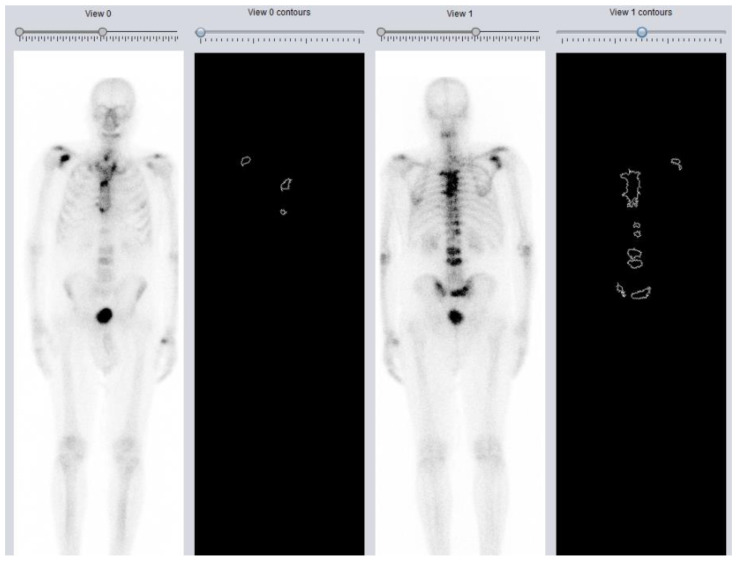
The 99mTc-HDP BS, an example of evaluation of the load of metastatic disease through DASciS software. BSI = 1.56%.

**Figure 2 biomedicines-11-01103-f002:**
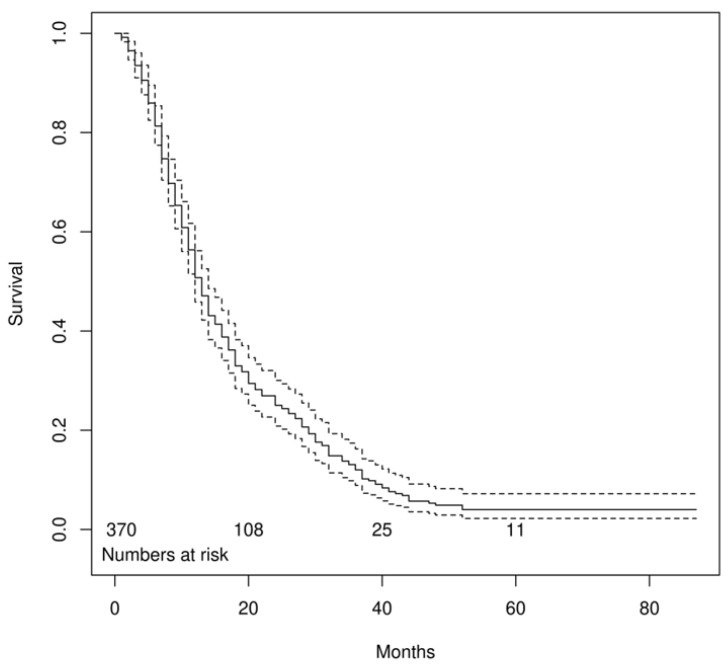
Kaplan-Meier curve showing overall survival in our cohort, with a 95% confidence interval in dashed lines (Median survival 13 months, CI: 12–14).

**Table 1 biomedicines-11-01103-t001:** Baseline patients’ characteristics (n = 370).

Variable	Mean (SD)	Patients	Percentage
Age (years)	73.6 ± 8.1		
Gleason score	7.9 ± 1.0		
5		2	0.5%
6		18	4.9%
7		105	28.4%
8		83	22.4%
9		98	26.5%
10		7	1.9%
unknown		57	15.4%
ECOG PS	0.7 ± 0.8		
0		176	47.6%
1		131	35.4%
2		58	15.7%
3		5	1.3%
Previous primary treatment			
Radical prostatectomy/Radiotherapy		159	43%
No		202	54.6%
missing		9	2.4%
Lymphadenopathies			
Yes		129	34.9%
No		203	54.8%
unknown		38	10.3%
Prior chemotherapy			
Yes		230	62.2%
No		140	37.8%
Use of bisphosphonates/denosumab			
Yes		193	52.2%
No		177	47.8%
Number of previous treatments	1.9 ± 1.4		
0		51	13.8%
1		102	27.5%
2		89	24.1%
≥3		119	32.2%
missing		9	2.4%
Number of cycles of ^223^RaCl_2_ received	5.3 ± 1.3		
1		8	2.2%
2		13	3.5%
3		26	7%
4		28	7.6%
5		33	8.9%
6		262	70.8%
Baseline tALP (U/L)	251.1 ± 310.9		
Baseline PSA (ng/ml)	230.7 ± 580.1		
Baseline Hb (g/dL)	11.9 ± 1.6		
Baseline BSI (%)	2.98 ± 2.42		

**Table 2 biomedicines-11-01103-t002:** Center-adjusted univariate—cox regression analysis of overall survival (OS) in relation to baseline variables.

Baseline Clinical Variables	HR	CI. Low	CI. Up	*p*-Value
Age	1.011	1.001	1.021	0.026
Radical prostatectomy/Radiotherapy	0.787	0.724	0.856	<0.001
Gleason score	1.115	1.016	1.225	0.022
Lymphadenopathies	1.437	1.014	2.037	0.042
ECOG	1.580	1.186	2.104	0.002
Hb	0.706	0.676	0.738	<0.001
tALP	1.001	1.001	1.001	<0.001
PSA	1.001	1.000	1.001	<0.001
BSI	1.137	1.052	1.230	0.001

**Table 3 biomedicines-11-01103-t003:** Center-adjusted multivariate—cox regression analysis of overall survival (OS) in relation to baseline variables.

Baseline Clinical Variables	HR	CI. Low	CI. Up	*p*-Value
Gleason score	1.096	1.051	1.144	<0.001
Hb	0.744	0.723	0.766	<0.001
tALP	1.001	1.000	1.001	<0.001
PSA	1.000	1.000	1.001	<0.001
BSI	1.054	1.040	1.068	<0.001

## Data Availability

The data presented in this study are available on request from the corresponding author.

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
