# Peer review of "The DASciS Software for BSI Calculation as a Valuable Prognostic Tool in mCRPC Treated with 223RaCl2: A Multicenter Italian Study"

_biomedicines, 2023, doi:10.3390/biomedicines11041103_

Round 1

Reviewer 1 Report

In this article, Maria Silvia De Feo et al. present the DASciS software for BSI calculation and show that it is a valuable prognostic tool in mCRPC treated with 223RaCl2. The data stem from a multicenter Italian study.

General comment:

The introduction of new software for BSI calculation and the results of the study are valuable. The paper is well written. The limitations of the study are fairly presented.

Minor points:

Line 180: 'AIC' abbreviation should be explained

Figure 2: Number at risk could be added.

Author Response

First of all, we would like to thank you for your comments and consideration.

According to your suggestions:

- we have explained the abbreviation AIC (Akaike Information Criterion) in “Statistical analysis”;

- we have added the numbers at risk in Fig. 2. 

Reviewer 2 Report

How is lymphadenopathy assessed by CT? If so, what was the bone finding on the CT scan, did it match the scintigraphy findings? How many patients had a CT scan?

Table 1.

Previous primary treatment Radical prostatectomy/Radiotherapy No missing 159 202 9 43% 54.6% 2.4%.

Unclear Radical prostatectomy/Radiotherapy – which number are matching?

Author Response

First of all, I would like to thank you for the attention shown and the precious suggestions made which we have welcomed. We have tried to clarify some data as follows.

 - We have specified in the manuscript that the presence of lymphadenopatis was assessed through a CT and/or a PET-CT by adding the following sentence in Materials and Methods, line 135: “…..assessed through a Computed Tomography (CT) scan and/or a Positron Emission Tomography (PET)-CT scan performed before enrollment...“.

Unfortunately, we do not have data concerning bone findings on CT scans. All patients had a CT and/or a PET-CT performed before enrollment to 223RaCl2 therapy to exclude the presence of visceral metastasis or malignant lymphadenopathies with more than 3 cm in the short-axis diameter; we do not have the specific data concerning how many patients had a CT scan rather than a PET-CT scan.

- We have correctly matched the variable with the corresponding value for the variable“ Previous primary treatment” in Table 1. Specifically:

- Radical prostatectomy/Radiotherapy 159 (43%)

- No 202 (54.6%)

- missing 9 (2.4%)

Reviewer 3 Report

There are various comments to be considered by authors, before the recommendation of acceptance. Therefore, a major revision is recommended.

Comment 1. Main text, check carefully whether it is 223RaCl2 or 223RaCl2.
Comment 2. Abstract:
(a) Check to ensure proper spacing, e.g., “95%CI”
(b) Enhance the discussion to highlight the key research results/findings.
Comment 3. Keywords, include more terms to better reflect the scopes of the paper.
Comment 4. Enhance the organization. Avoid present with lengthy paragraph.
Comment 5. Section 1 Introduction:
(a) Check to ensure proper spacing, e.g., between wordings and in text citations [x].
(b) Literature review is incomplete. Please include a summary of the methodology, results, and limitations of existing works.
(c) Summarize the research contributions.
Comment 6. Section 2 Materials and Methods:
(a) The data collection process must be elaborated.
(b) The description of the method is too simple. Make elaboration on the technical details/formulations.
Comment 7. Section 3 Results:
(a) Check for the formats of figures and tables using the journal’s template.
(b) Ensure sufficient elaboration and explanation are shared for all tables and figures.
Comment 8. Compare the results with existing works.

Author Response

First of all, I would like to thank you for the precious suggestions made which we have welcomed. We tried to put them in place as follows.

Comment 1

We have used the correct form 223RaCl2 in the whole manuscript.

Comment 2. Abstract

a) As specified, we have ensured the proper spacing of “95%CI”

  1. b) According to your suggestion, we added the following sentence to better clarify the statistical results “….meaning that patients with higher BSI values had worse OS”.

Comment 3

Following your suggestion we have added more terms in the keywords, specifically overall survival, bone metastasis, and bone scintigraphy

Comment 4

According to your suggestion, we have divided the Introduction in four paragraphs as follows:

1) 1.1. Radium-223 dichloride (223RaCl2)

2) 1.2. Bone scan index (BSI)

3) 1.3. DASciS software for BSI calculation

4) 1.4. Aim of the study

Comment 5. Section 1 Introduction:

According to your suggestions:

  1. a) we have ensured the proper spacing between words and in the citations;
  2. b) we have added a mention on the two most common software developed for BSI calculation in the final part of the Introduction with the corresponding references as follows “ The BONENAVI® software (Japan), as well as the EXINI bone software package (EXINI Diagnostics, Lund, Sweden) have been specifically developed for BSI calculation, enabling the use of this prognostic measure in prostate cancer patients”.

Comment 6. Section 2 Materials and Methods

  1. a) According to your suggestion we specified the data collection process as follows “Other baseline clinical variables relevant for OS analysis specifically age, Gleason Score, Eastern Cooperative Oncology Group (ECOG) performance status (PS) score, previous primary treatment, presence of lymphadenopathies, prior chemotherapy, use of bisphosphonates/denosumab, number of previous treatments, number of cycles of 223RaCl2 received, baseline values of hemoglobin (Hb), total alkaline phosphatase (tALP) and prostate specific antigen (PSA), were independently collected by the six involved centers and then shared, put together and taken into account for the statistical analysis.”
  2. b) Following your precious consideration, we have added some details concerning the calculation of BSI as follows: “The DASciS software developed by the engineering team from Sapienza University of Rome was shared with all involved Nuclear Medicine Units. After an online training, the six centers independently processed the baseline BS images of their patients with the DASciS software. BSI values were independently collected and then put together for the statistical analysis.”

Comment 7. Section 3 Results:

According to your considerations:

  1. a) we have checked the formats of figures and tables using the journal’s template
  2. b) To better ensure sufficient explanation of figures, we added the numbers at risk in figure 2 and we correctly matched some variables with the corresponding values in Table 1.

Comment 8.

Following your consideration we have added the following sentence (and the corresponding references) to make a comparison with the results obtained with other software for BSI calculation. “Our results are in line with data reported in the abovementioned papers and obtained by using different software for reliable BSI calculation”

Reviewer 4 Report

Chapter 1: Introduction.

Please use Paragraphs (at least 3), so that the content can be easily followed by readers

Chapter 2: Methodology

The Cluster AI algorithm to determine the baseline BSI essentially assesses Disease Burden. We know that patient with a High Burden of disease have a worse prognosis, as seen in CHAARTED trial. Hence the assumption that the prognostic value of the BSI score is only relevant to response to RA223 is abstract. The patients with a high BSI score had a worse prognosis due to the volume of disease, rather than treatment choice. The BSI score is useful, but as a prognostic biomarker, rather than a predictive biomarker.

Chapter 3: Results.

Line 192: 'Approximately 1/3 of patients had lymphadenopathy'. The EMA licence for RA223 is for bone 'only' disease. If these patients had significant volume of lymphadenopathy they would not be eligible for Radium treatment according to NICE.

Line 211: Gleason Score, PSA and ALP are independent prognostic indicators as well. 

Chapter 4 Discussion:

Please highlight that this is a hypothesis generating trial and not a proof of principle. As I mentioned previously the BSI score calculated disease burden and this is an independent prognostic indicator for OS. Also the multivariate analysis confirms that Gleason score, HB, ALP and other clinical parameters carry a prognostic value. We need to accept that this Cluster AI software is useful and these findings are important, but should be validated.

Author Response

First of all, I would like to thank you for the attention shown and the precious suggestions made which we have welcomed. We tried to put them in place as follows.

Chapter 1: Introduction

We have used four paragraphs to make the content clearer for readers:

1) 1.1. Radium-223 dichloride (223RaCl2)

2) 1.2. Bone scan index (BSI)

3) 1.3. DASciS software for BSI calculation

4) 1.4. Aim of the study

Chapter 2: Methodology and Chapter 4: Discussion

According to your precious consideration, we have clarified that the BSI represents a prognostic biomarker as patients with a high burden of disease have a worse prognosis independently from treatment choice by adding the following sentences:

- This measure has proven to be a valuable prognostic marker in patients with prostate cancer, allowing the identification of subjects with distinct prognoses for better stratification in clinical trials, as its prognostic significance is independent of treatment choice.

- Based on these considerations and due to its valuable utility in patients’ stratification, independently from treatment choice, baseline BSI could be investigated as an additional variable to be included in a multidimensional approach taking into account different data, such as baseline Hb, tALP and PSA which corroborated their prognostic relevance in the present analysis.

In addition, we have replaced the word predictive with the word prognostic in the end of the Introduction.

Chapter 3: Results.

Line 192: we have specified in Materials and Methods that all patients had a CT and/or a PET-CT performed before enrollment to exclude the presence of visceral metastasis or malignant lymphadenopathies with more than 3 cm in the short-axis diameter, to better underline that the presence of lymphadenopathies with less than 3 cm in the short-axis diameter do not represent an exclusion criterion for 223RaCl2 therapy

Line 211: The value of other independent prognostic factors in addition to BSI is reported in the following sentence: “Concerning the other clinical variables exhibiting a significant association with OS, higher age, Gleason score, ECOG-PS, tALP, and PSA, as well as the presence of lymphadenopathies at the time of enrollment, resulted independently associated with an increased risk of death, while previous primary treatment (radical prostatectomy/radiotherapy) and higher baseline Hb values significantly associated with better outcomes.”.

We have avoided writing all the corresponding statistical values as they are extensively reported in Table 2.

Round 2

Reviewer 3 Report

I have some minor follow-up comments.
Follow-up comment 1: Consider updating "95%CI" to "95% CI" with a spacing.

Follow-up comment 2: Ensure high resolutions for all figures, and enlarge the WORD/PDF file to confirm.

Follow-up comment 3: Make elaboration on future research directions.

Author Response

I would like to thank you again for the attention shown. According to your suggestions we have made the following changes:

- Follow-up comment 1: we have updated "95%CI" to "95% CI" in the whole manuscript;

- Follow-up comment 2: we have ensured high resolution for all figures and enlarged the WORD/PDF file;

- Follow-up comment 3: we have added the following sentence concerning future research direction at the end of the Discussion: “An interesting future step of our research could be represented by using the DASciS software for the calculation of BSI changes after three cycles of 223RaCl2 and at the end of treatment, aiming to assess whether patients with longer OS also show a decline of bone metastatic load at BS.”

Reviewer 4 Report

Thank you for adopting suggestions for improving the manuscript

Author Response

We would like to thank you again for the precious suggestions received.